# Host Jumps and Pathogenicity of Botryosphaeriaceae Species on Grapevines (*Vitis vinifera*) in Chile

**DOI:** 10.3390/microorganisms13020331

**Published:** 2025-02-03

**Authors:** Yadira Hernández, Mauricio Lolas, Karina Elfar, Akif Eskalen, Felipe Gainza-Cortés, Gonzalo A. Díaz

**Affiliations:** 1Laboratory of Fruit Pathology, Faculty of Agricultural Sciences, University of Talca, Talca 3460000, Chile; yadira.hernandez@utalca.cl (Y.H.); mlolas@utalca.cl (M.L.); 2Department of Plant Pathology, University of California, Davis, CA 95616, USA; kdelfar@ucdavis.edu (K.E.); aeskalen@ucdavis.edu (A.E.); 3Viña Concha y Toro S.A., Center for Research and Innovation, Fundo Pocoa s/n, Km10 Ruta K-650, Pencahue 3550000, Chile; felipe.gainza@conchaytoro.cl

**Keywords:** cross pathogenicity, Botryosphaeriaceae, grapevine dieback, aggressivity, fungal trunk pathogens

## Abstract

Botryosphaeria dieback disease is a significant grapevine trunk disease (GTD) caused by species of Botryosphaeriaceae in Chile and worldwide. Moreover, Botryosphaeriaceae have been described attacking fruit and nut crops in Chile. However, it remains unknown whether fungal isolates from tree hosts have the potential to infect grapevines in Chile. The aggressiveness and potential cross infection of species of Botryosphaeriaceae collected from crops (grapevines, apples, blueberries, and walnuts), was assessed on grapevines. Plant materials, including nursery cuttings, lignified canes, and green shoots of grapevine cultivars (Cabernet Sauvignon, Syrah, Sauvignon Blanc, Malbec, Aspirant Bouschet, and Merlot), were inoculated with *Diplodia mutila*, *D. seriata*, *Dothiorella sarmentorum*, *Lasiodiplodia theobromae*, *Neofusicoccum arbuti*, and *N. parvum*, under greenhouse and vineyard conditions. Regardless of the origin of the isolates, most species of Botryosphaeriaceae were pathogenic on plant materials, causing necrotic lesions of mean lengths from 11.9 to 86.2 mm using mycelial suspension and from 24.8 to 253.7 mm with mycelial plugs. Notably, *Neofusicoccum* species were the most aggressive, regardless of host origin. Other less aggressive species included *L. theobromae*, *D. mutila*, and *D. seriata* isolated from apple and walnut. This study highlights the potential of species of Botryosphaeriaceae from alternative fruit hosts as inoculum sources for grapevines in Chile.

## 1. Introduction

The Botryosphaeriaceae Theiss. and Syd. family (Ascomycota: Botryosphaeriales) includes a wide range of fungal species, with at least 85 well-defined genera [1]. These fungi can act as endophytes, saprophytes, and plant pathogens with cosmopolitan distribution [2,3,4]. Several species of Botryosphaeriaceae are aggressive plant pathogens, responsible for a variety of diseases, including leaf spots, fruit rots, dieback, and perennial cankers in fruit trees, vines, forest, and urban plants [5,6,7,8,9].

One of the most destructive diseases affecting grapevines (*Vitis vinifera* L.) is Botryosphaeria dieback, a grapevine trunk disease (GTD) that causes severe economic losses worldwide [10,11,12]. This disease has been reported in Australia [13], California [14], Chile [15], China [16], New Zealand [17], Italy [6], Portugal [18], South Africa [19], and Spain [20,21], among others.

In Chile, grapevine is a major crop, with a wine-growing area of 139,179 ha. Of this area, 74% is dedicated to red grape cultivars, with Cabernet Sauvignon planted most at 40,053 ha, and the 26% of white grape varieties dominated by Sauvignon Blanc [22]. Over the past decade, numerous studies have revealed a considerable prevalence of decline and dieback symptoms in vineyards and commercial orchards. Species of Botryosphaeriaceae have been isolated and identified from the canes, spurs, and cordons of grapevines [23,24]. Significant economic damage has been reported associated with Botryosphaeria dieback, with disease prevalence ranging from 15 to 85% [15,24]. Therefore, this disease is a limiting factor for fruit and wine production and exports from central Chile [24].

The broad host range of species of Botryosphaeriaceae is well-documented internationally [25,26,27,28]. Only a few studies [29,30] have explored the potential of inoculum sources from other fruit host species. In Chile, species of the Botryosphaeriaceae have been isolated and identified from various other crops, including blueberry [31], walnut [32], apple [33], and avocado [34]. Given this host range and their proximity to each other in Chilean production areas, research on cross infection between host species is highly relevant, especially since these fungi are also reported in multiple species cultivated near each other in regions like Uruguay, California, South Africa, New Zealand, and Spain [11,12,23,30].

These background findings, together with prior evidence on Botryosphaeriaceae species affecting various hosts, support the hypothesis of cross infection between non-grapevine hosts and grapevines. These antecedents underscore the relevance of conducting studies to demonstrate cross infection between different fruit tree species and grapevines, particularly in Chile. Further research is necessary to clarify the role of inoculum from non-grapevine hosts surrounding vineyards in the epidemiological cycle of diseases caused by Botryosphaeriaceae species. This information will be critical in guiding future studies aimed at improving the management of these diseases. Therefore, the present study aimed to determine the aggressiveness of species of Botryosphaeriaceae isolates collected from different fruit tree hosts (apple, blueberry, walnut, and grapevines) on different tissues of grapevine cultivars under Mediterranean conditions in the Maule Region of central Chile.

## 2. Materials and Methods

### 2.1. Fungal Isolates

Thirteen fungal isolates representing six species of the Botryosphaeriaceae family, namely *Diplodia mutila*, *D. seriata*, *Dothiorella sarmentorum*, *Lasiodiplodia theobromae*, *Neofusicoccum arbuti*, and *N. parvum*, were isolated from branches or cordons with canker and dieback symptoms from various fruit tree species (apple, blueberry, grapevine, and walnut) located in commercial orchards in the Maule Region, Central Chile, obtained from this and previous studies (Table 1). The isolates were cultured and maintained on 2% potato dextrose agar (PDA) (Merck S.A., Darmstadt, Germany) and incubated at 20 °C under a 12 h light/dark photoperiod.

### 2.2. Plant Material and Location

For greenhouse studies, lignified grapevine canes cvs. Cabernet Sauvignon, Syrah, Sauvignon Blanc, Malbec, and Aspirant Bouschet were obtained from Concha y Toro vineyard, Center for Research and Innovation, Fundo Pocoa, Pencahue, (35°03′48″ S 71°16′19″ O), Maule Region, Chile. Dormant canes (50 cm, 1 year old) were collected during the winter of 2021 and 2022. The cuttings were superficially disinfected by immersion in 0.5% chlorine bleach for 1 min, followed by a 3 min rinse in sterile distilled water, and left to dry on absorbent paper at room temperature (20 to 22 °C) for 30 min. Once dry, they were dipped in a commercial rooting agent (*Estimulante Enraizante*, Anasac, Chile) for 30 s and placed vertically in sterile perlite beds in 60 × 30 cm plastic boxes, previously disinfected. The sterile perlite beds were prepared by adding 10 L of perlite to boxes and moistening it with 3 L of water. Ten cuttings were inserted 5 cm deep into the moistened perlite, and inoculated the following day. The cuttings were maintained under controlled greenhouse conditions with temperatures from 18 to 24 °C, relative humidity levels of 47 to 85%, 14 h light/10 h dark cycle, and watered 1 to 3 times per week as needed. The greenhouse was located at the Fruit Pathology Laboratory, Talca Campus, University of Talca (35°24′ S, 71°37′ W).

Field studies were conducted at the El Llano vineyard using cvs. Cabernet Sauvignon and Merlot, which were trained as bilateral cordons, under a drip irrigation system, located in San Clemente (35°33′ S, 71°28′ W), Maule Region, Central Chile.

### 2.3. Mycelial Suspension Inoculation Method

Mycelial suspensions were prepared by adding 3 mL of 0.1% Tween 80 (Sigma-Aldrich, St. Louis, MO, USA) to Petri dishes with 7-day-old mycelial cultures. The surface of the media was gently scraped with a sterile scalpel and fragments were ground using a blender (Moulinex LM233A, Alençon, Francia) (1 min, speed 5), then the mixture was filtered through two layers of cheesecloth and adjusted to a concentration of 10^5^ mycelial fragments/mL using a hemocytometer (Precicolor, HGB, Gießen, Germany) [36]. The suspensions were prepared 12 h before inoculation and stored at 4 °C until use.

In the greenhouse, Cabernet Sauvignon and Syrah cuttings were pruned using alcohol-disinfested pruning shears at an angle of 45° and 3 cm above in the distal zone and immediately inoculated with 40 μL of mycelial suspension using a micropipette [37]. Sterile distilled water was used for the control. After inoculation, cuttings were maintained for 4 months. The experiment was repeated.

For the field study, four rows of cv. Cabernet Sauvignon were cane-pruned to five buds during the dormancy stage (July) using alcohol-disinfested pruning shears at an angle of 45° and 2–3 cm above the five buds in each cane [20]. Fresh pruning wounds were immediately inoculated with a 40 µL drop of a mycelial suspension (10^5^ fragments/mL) placed with a micropipette on each pruning wound [37]. Sterile distilled water was used for the untreated control. Canes were collected for laboratory analysis 5 months after inoculation.

### 2.4. Mycelial Plug Inoculation Method

In the greenhouse, cuttings of Cabernet Sauvignon, Syrah, Sauvignon Blanc, Malbec, and Aspirant Bouschet were wounded in the internode area with a scalpel to create a wedge-shaped lesion reaching the xylem tissue without penetrating the pith. Fresh mycelial plugs (5 mm in diameter) were then inserted into the wounds, with the mycelium side facing inward, and the inoculated wounds were wrapped with Parafilm (Pechiney Plastic Packaging, Chicago, IL, USA) to prevent desiccation. After inoculation, cuttings were maintained for 6 months. The experiment was repeated.

For field studies, four rows of cv. Cabernet Sauvignon and cv. Merlot, with 10 attached green shoots each, were inoculated in December 2022, using the same methodology as described above. Attached green shoots were collected for laboratory analysis 8 months after inoculation. This experiment was repeated once.

For evaluation in both inoculation methods, the cuttings (greenhouse), canes, and green shoots attached (vineyards), were collected and analyzed in the laboratory. Necrotic lesions were measured with a digital caliper, for pruning wound cuts, measured from wound to edge of necrosis, and for internode wedge wounds, measured as total necrosis both upwards and downwards from wound. Data from repeated experiments were combined for analysis.

### 2.5. Reisolated

To confirm Koch’s postulates, fungi were reisolated from wood tissue taken from the edge of lesions. The wood surface was disinfected in 2.5% sodium hypochlorite (Clorinda, Clorox Chile S.A., Santiago, Chile) for 10 min and then washed twice. Using flame-sterilized secateurs, small wood pieces (approximately 3 × 2 × 2 mm) covering the interface between normal and discolored tissue were cut and placed on PDA medium and reidentified based on their cultural and morphological characteristics [14,33].

### 2.6. Data Analysis

The mycelial suspension inoculation experiments in the greenhouse were carried out according to a 2 × 10 (cultivars × isolates) factorial design, with 3 replicates, each of 15 cuttings per box as the experimental unit. The field experiment was performed in a completely randomized design, with 4 replicates each of 10 canes in two consecutive vines as the experimental units.

The mycelial plug inoculation experiments in the greenhouse used a 5 × 12 (cultivars × isolates) factorial design, with 3 replicates, each of 15 cuttings per box as the experimental unit. The field experiments were performed according to a 2 × 12 (cultivars × isolates) factorial design, with 3 replicates each of 10 attached green shoots in two consecutive vines as the experimental units.

Lesion lengths were analyzed as the dependent variable for both inoculation methodologies using analysis of variance (ANOVA). The Shapiro–Wilk and Levene’s tests were used to verify normality and homogeneity of variances, respectively, across all data grouped by the two factors. When necessary, data were log-transformed (base 10) to meet these assumptions. Tukey’s multiple range post hoc test was applied to compare mean lesion lengths across isolates and cultivars. Data processing was conducted in R Studio 4.1.0.

## 3. Results

### 3.1. Mycelial Suspension Inoculation Method

Under greenhouse conditions, all Botryosphaeriaceae isolates caused symptoms of necrotic lesions and dieback, varying in severity. These symptoms included dark discoloration and internal wood streaking, indicative of pathogen colonization. In the vineyard, similar necrotic lesions and dieback were observed on inoculated canes of Cabernet Sauvignon.

Under greenhouse condition, a significant interaction effect between cultivar and isolate was observed (*p* < 0.001). In the cv. Cabernet Sauvignon, the most extensive lesions were caused by the isolates of *N. parvum* from grapevine (Np-vine), *N. arbuti* from apple (Na-apple), from walnut *D. mutila* (Dm-wal1), *N. parvum* (Np-wal), and *N. parvum* from blueberry (Np-blue1) (Table 2, with the lowercase letter “a” identifying the most extensive lesions mean within this cultivar). Mean lesion lengths ranged from 34.7 mm (Np-vine) to 28.1 mm (Np-blue1). In the cv. Syrah, the most extensive lesions mean were caused by *N. parvum* from grapevine (Np-vine) and *N. arbuti* from apple (Na-apple) (Table 2, with the lowercase letter “d” identifying the most extensive lesion means within this cultivar), with mean lesion lengths ranging from 21.4 to 21.3 mm, respectively. Despite sharing the same isolates responsible for the highest mean lesion lengths, the magnitude of these lesions differed significantly between the two cultivars. The interaction between cultivar and isolate was evident not only for the most aggressive isolates but also for all others, except for *D. mutila* from apple (Dm-apple), which differed significantly from each other.

The cultivar had a significant effect on necrotic lesion length (*p* < 0.001). Overall, the cv. Cabernet Sauvignon exhibited significantly greater susceptibility compared to the Syrah, with mean lesion lengths of 24.9 and 16.0 mm, respectively (Table 2).

Isolates also had a significant effect on lesion development (*p* < 0.001), with mean lesion lengths varying widely across isolates. *N. parvum* from grapevine (Np-vine; 28.0 mm) was the most aggressive isolate across cultivars, followed by *N. arbuti* from apple (Na-apple; 26.3 mm), N. parvum from walnut (Np-wal; 24.8 mm), and N. parvum from blueberry (Np-blue1; 24.3 mm). These isolates were significantly more aggressive than the others. Among the least aggressive were *D. seriata* from apple (Ds-apple; 20.0 mm), *D. mutila* from apple (Dm-apple; 15.8 mm), and *L. theobromae* from apple (Lt-apple; 15.5 mm). All isolates differed significantly from the control (Table 2). The fungal species were successfully reisolated from the discolored tissues of all inoculated cuttings, thereby fulfilling Koch’s postulates, while the controls consistently yielded negative results.

Under vineyard conditions, mean lesion lengths varied significantly between Botryosphaeriaceae isolates on Cabernet Sauvignon (Table 2). Overall, *N. parvum* (Np-wal) and *D. mutila* (Dm-wal1) from walnut were the most aggressive isolates, with mean lesion lengths of 86.2 and 84.6 mm, respectively. Isolates of *N. parvum* from blueberry (Np-blue1 and Np-blue2) also exhibited considerable virulence, with lesion lengths of 61.4 and 73.4 mm, respectively. In contrast, *L. theobromae* (Lt-apple) and *D. seriata* (Ds-apple) from apple were the least aggressive, causing lesions of 24.5 and 38.0 mm, respectively (Table 2, lowercase letters). The fungal species were successfully reisolated from the discolored tissues of all inoculated canes, thereby fulfilling Koch’s postulates, while the controls consistently yielded negative results.

### 3.2. Mycelial Plug Inoculation Method

In the greenhouse, the grapevine cuttings of the five cultivars (Cabernet Sauvignon, Syrah, Sauvignon Blanc, Malbec, and Aspirant Bouschet) inoculated with the species of Botryosphaeriaceae developed visible necrotic lesions extending both upwards and downwards from the wedge-shaped wounds (Figure 1A–E). In the vineyard, inoculated green shoots of Cabernet Sauvignon and Merlot also showed necrotic lesions spreading in both directions from the wedge-shaped wounds (Figure 1F–G). These symptoms included dark discoloration and internal wood streaking, indicative of pathogen colonization.

Under greenhouse conditions, a significant interaction effect between cultivar and isolate was observed (*p* < 0.001). In the cv. Cabernet Sauvignon, the most extensive lesions were caused by the isolates *N. parvum* from walnut (Np-wal), *N. parvum* from grapevine (Np-vine), *N. arbuti* from apple (Na-apple), *N. parvum* from blueberry (Np-blue1), *D. mutila* from apple (Dm-apple), and *D. seriata* from grapevine (Ds-vine), with mean lesion lengths ranging from 253.7 mm (Np-wal) to 112.6 mm (Ds-vine). In cv. Syrah, the most extensive mean lesion lengths were associated with the isolate *N. parvum* from blueberry (Np-blue1; 103.4 mm). In cv. Malbec, the most extensive mean lesion lengths were also associated with the isolate *N. parvum* from blueberry (Np-blue1; 196.6 mm). In cv. Sauvignon Blanc, the most extensive mean lesion lengths were associated with the isolate *N. parvum* from grapevine (Np-wine; 142.2 mm). In cv. Aspirant Bouschet, the most extensive mean lesions lengths were associated with the isolates *N. parvum* from walnut (Np-wal) and *D. seriata* from apple (Ds-apple), with lengths of 174.9 and 134.9 mm, respectively (Table 3, where the inclusion of lowercase “a” identifies the most extensive lesion mean within each cultivar). While some isolates associated with the most extensive lesions were shared among certain cultivars, as mentioned above, the diversity of isolate-cultivar combinations emphasizes the complexity of the interaction. For example, the isolate *D. seriata* from grapevine (Ds-vine) caused a mean lesion of 112.6 mm in Cabernet Sauvignon, ranking among the most aggressive in this cultivar, while in Aspirant Bouschet, the lesion caused by the same isolate was significantly smaller at 32.8 mm, demonstrating lower aggressiveness. Another interaction example corresponds to *N. parvum* from walnut (Np-wal), producing the longest mean lesion observed across all cultivar-isolate combinations in Cabernet Sauvignon and the most extensive lesion among isolates associated with Aspirant Bouschet. This isolate produced reduced lesion lengths in Syrah, Malbec, and Sauvignon Blanc, with 93.4, 69.3, and 76.0 mm, respectively, which were significantly smaller than the most extensive lesions in Cabernet Sauvignon but not significantly different from the lesions in Aspirant Bouschet (see details of the varied interactions represented by lowercase letters in Table 3).

The cultivar had a significant effect on necrotic lesion length (*p* < 0.001). Cabernet Sauvignon was the most susceptible cultivar, followed by Malbec, Aspirant Bouschet, Sauvignon Blanc, and Syrah. Notably, Cabernet Sauvignon exhibited significantly larger necrotic lesions than the other cultivars, which did not significantly differ from each other (Table 3).

Isolates also had a significant effect on lesion development (*p* < 0.001), with lesion lengths varying widely across isolates. Isolates from *Neofusicoccum* species were the most aggressive across cultivars and regardless of the original host (Table 3, where the inclusion of letter “a” identifies the most extensive mean lesions across isolates) ranging from *N. parvum* from walnut (Np-wal; 133.5 mm) to *N. arbuti* from apple (Na-apple; 97.1 mm). These isolates were significantly more aggressive than the others. Among the least aggressive were *D. mutila* from walnut (Dm-wal1; 38.9 mm), *L. theobromae* from apple (Lt-apple; 40.0 mm), and *Do. sarmentorum* from walnut (Dsar-wal; 44.9 mm). All isolates differed significantly from the control (Table 3). The fungal species were successfully reisolated from the discolored tissues of all inoculated cuttings, thereby fulfilling Koch’s postulates, while the controls consistently yielded negative results.

Under vineyard conditions, the isolate × cultivar interaction was significant (*p* < 0.001). In the cv. Cabernet Sauvignon, the most extensive mean lesions were associated with the isolates *N. parvum* from blueberry (Np-blue1), *N. parvum* from walnut (Np-wal), and *N. parvum* from grapevine (Np-vine), with mean lesion lengths ranging from 151.6 mm (Np-blue1) to 95.9 mm (Np-vine). In contrast, for cv. Merlot, the isolates *N. parvum* from grapevine (Np-vine) and *N. parvum* from walnut (Np-wal) also caused the most extensive mean lesions, measuring 114.2 and 113.8 mm, respectively (Table 3, the lowercase letter “a” identifies the most extensive mean lesions for each cultivar). Notably, *N. parvum* from blueberry (Np-blue1), while being one of the most aggressive isolates in Cabernet Sauvignon, showed a significant difference between the two cultivars, with lesion lengths of 151.6 mm in Cabernet Sauvignon and 94.3 mm in Merlot, further emphasizing the differential aggressiveness of this isolate between cultivars. Additionally, other isolates, such as *D. mutila* from walnut (Dm-wal1), *D. seriata* from apple (Ds-apple), and *L. theobromae* from apple (Lt-apple) exhibited cultivar-specific differences in mean lesion lengths, demonstrating variable levels of aggressiveness across different cultivars (Table 3, lowercase letters).

Both cultivar and isolate had a significant effect (*p* < 0.001) on necrotic mean lesion length. Based on the means of lesion diameters caused by the 12 isolates of Botryosphaeriaceae tested, Cabernet Sauvignon was significantly more susceptible than Merlot (Table 3).

The isolates of *N. parvum* were the most virulent across both cultivars and regardless of the original host. Mean lesion lengths caused by *N. parvum* ranged from 95.9 mm (Np-vine) to 151.6 mm (Np-blue1) in Cabernet Sauvignon, and from 94.3 mm (Np-blue1) to 114.2 mm (Np-vine) in Merlot. The mean lesion lengths caused by *N. parvum* isolates (Np-blue1; 123.0 mm, Np-wal; 124.0 mm, and Np-vine; 105.0 mm) were significantly greater than those caused by isolates of other species, highlighting their superior aggressiveness. Among the least aggressive were *D. seriata* from apple (Ds-apple; 43.6 mm), from grapevine (Ds-vine; 51.5 mm), and *L. theobromae* from apple (Lt-apple; 55.3 mm). All isolates differed significantly from the control (Table 3). The fungal species were successfully reisolated from the discolored tissues of all inoculated green shoots, thereby fulfilling Koch’s postulates, while the controls consistently yielded negative results.

## 4. Discussion

The present research is the first study demonstrating that species of Botryosphaeriaceae causing canker and dieback in other fruit tree hosts (apple, blueberry, and walnut) are pathogenic on lignified tissues of grapevines, and these fungal trunk pathogens are potential sources of inoculum for grapevines, especially when the fruit crops are cultivated in proximity to commercial vineyards, as in Central Chile.

This study demonstrated that almost all evaluated species of Botryosphaeriaceae, regardless of the isolate or original host and despite the inoculation method used (mycelial suspension and mycelial plug), were capable of infecting, colonizing, and causing necrotic lesions on lignified grapevine cuttings, green shoots, and lignified canes across all tested cultivars, under both greenhouse and vineyard conditions, as evidenced by consistent significant differences compared to control treatments.

These findings provide important insights into cross infections, demonstrating that species of Botryosphaeriaceae obtained from various fruit tree hosts can cause necrotic lesions on multiple grapevine cultivars. This confirms that their original host does not restrict the pathogenic potential of these fungal isolates, as they can infect a range of grapevine cultivars under both controlled and field conditions.

Our results align with those of Mojeremane et al. [20] in South Africa, which demonstrated that *Neofusicoccum* isolates (*N. australe* and *N. stellenboschiana*), sourced from a variety of hosts including grapevine, fig, plum, olive, apple, and Peruvian pepper, were pathogenic to grapevines in laboratory trials. Similarly, Cloete et al. [38] showed that Botryosphaeriaceae species like *N. vitifusiforme*, *N. australe*, *Diplodia* sp., and *D. seriata,* isolated from apple and pear trees with dieback symptoms, were pathogenic to grapevine cuttings in laboratory trials. Consistently, in New Zealand, Amponsah et al. [39] observed that species of Botryosphaeriaceae, including *N. luteum*, *N. parvum*, *N. australe*, *D. mutila*, and *D. seriata,* from hosts such as blueberry, juniper, willow, cherry, oak, lemon, pine, olive, apple, plum, and grapevine caused necrotic lesions on green grapevine shoots after three months of incubation. According to their findings, all isolates, except for *D. seriata,* caused lesions regardless of host of origin.

Our results indicate that isolates obtained from grapevine tend to be more aggressive, causing more extensive necrotic lesions. Specifically, *N. parvum* from grapevine (Np-vine) was among those that caused the most extensive lesions. This aligns with Van Dyk et al. [40], who reported that isolates obtained from the same host (European olive) exhibited greater aggressiveness than those from another host (wild olive), despite the latter isolates also being pathogenic. This highlights the complexity of cross infection and its impact on virulence. While isolates from the same host typically show higher virulence, this is not always consistent. For instance, *N. parvum* from walnut and *N. arbuti* from apple caused longer lesions in certain grapevine cultivars (Cabernet Sauvignon, Malbec y Aspirant Bouschet) than the *N. parvum* isolate from grapevine. In contrast, in our study, *D. seriata* from grapevine (Ds-wine) did not exhibit the same pattern, as it caused less extensive lesions compared to *N. parvum*, which may be due to differences in phytotoxic metabolite production, as *Neofusicoccum* species are known to produce more such metabolites than *D. seriata* [41].

The aggressivity of species of Botryosphaeriaceae varies within species and among isolates, exhibiting different levels of virulence. In this study, the most extensive necrotic lesions were most frequently caused by *Neofusicoccum* species, regardless of their host of origin. These results agree with previous studies that identified *Neofusicoccum* as highly aggressive. In this framework, several studies [38,39,42,43] reported *N. luteum*, *N. australe*, and *N. parvum* as the most aggressive in pathogenicity trials on green shoots and lignified grapevine cuttings, in laboratory, greenhouse, and field conditions. *Neofusicoccum* species are known to colonize wood more rapidly and are frequently detected in vineyards worldwide, including New Zealand [39], the United States [44], Spain [45], and Australia [46]. Pitt et al. [46] identified *N. parvum* and *L. theobromae* as the most virulent species in Chardonnay vineyards, which supports the aggressive behavior of *N. parvum* observed in our study.

According to Martos et al. [41], as mentioned above, the high virulence of *Neofusicoccum* could be attributed to the large number of phytotoxic metabolites it produces compared to other genera, such as *D. seriata*, which typically cause shorter lesions [14,39,47,48]. In this study, *D. seriata* and *D. mutila* showed moderate aggressiveness on grapevine cuttings, canes, and green shoots. The aggressivity of *D. mutila* has varied in studies; Phillips [49] in Portugal, and Taylor et al. [48] in Australia, described *D. mutila* as weakly aggressive, while Van Niekerk et al. [19] in South Africa, and Billones-Baaijens et al. [17] in New Zealand identified it as a more aggressive pathogen, suggesting it may act as either a primary or opportunistic pathogen, depending on environmental factors.

Regardless of inoculation method, mycelial suspension or plug, the least virulent isolates were *L. theobromae* (from apple), *D. mutila* (from walnut), and *D. seriata* (from apple). Although *L. theobromae* has been reported as a highly virulent species in various field studies conducted on shoots or cuttings [42,43,46], it has also shown low virulence in some in vitro assays [42]. This inconsistency may be attributed to environmental conditions affecting disease development. Temperature strongly influences its physiology and aggressiveness as a subtropical and tropical plant host species. Studies in Australia, Mexico, Portugal, and the United States have identified *L. theobromae* as highly aggressive in grapevines [13,18,43]. In Chile, *L. theobromae* has shown lower virulence compared to other regions, where it is more aggressive [33]. This difference can be attributed to the environmental conditions specific to Chile, particularly the climate. While *L. theobromae* is well-suited to subtropical environments, where it thrives and exhibits high virulence, in Chile, its distribution is more restricted to northern localities characterized by warmer and drier conditions. The Mediterranean climate of central Chile, with its cooler and more variable temperatures, may not provide the ideal conditions for *L. theobromae* to express its full pathogenic potential.

This demonstrates that environmental factors can influence the aggressiveness of different species of Botryosphaeriaceae. It has been reported that high levels of humidity and precipitation favor spore germination and the penetration of the fungus into plant tissues, particularly in tropical climates [50,51]. Furthermore, temperature can affect the production of secondary metabolites in these phytopathogens, which in turn influences their ability to degrade the cell walls of wood, thereby impacting their level of aggressiveness [12].

Regarding grapevine cultivar, Cabernet Sauvignon proved to be the most susceptible under both greenhouse and vineyard trials, across both inoculation methods, when comparing it to Syrah (greenhouse mycelial suspension) and Merlot (vineyard mycelial plug). Furthermore, Cabernet Sauvignon displayed significantly higher susceptibility than Malbec, Aspirant Bouschet, Sauvignon Blanc, and Syrah (greenhouse micellar plug), confirming its increased vulnerability.

In contrast, Amponsah et al. [39] in New Zealand evaluated the virulence of various Botryosphaeriaceae isolates on five grapevine cultivars—Cabernet Sauvignon, Pinot Noir, Sauvignon Blanc, Riesling, and Chardonnay, under greenhouse and controlled conditions and they found no significant differences among cultivars. All cultivars were equally susceptible to *N. luteum*, *N. parvum*, *N. australe*, and *D. mutila*. Similarly, Billones-Baaijens et al. [52] found that the six popular rootstocks and cultivars were susceptible to *N. luteum*, *N. parvum*, and *N. australe*, with Merlot standing out as one of the most susceptible, showing the longest necrotic lesions.

The differences in susceptibility among the grapevine cultivars may be linked to the variability in the diameter or thickness of their vessels. This was previously reported by Pouzoulet et al. [53], who reported smaller vessel diameter in less susceptible cultivars like Malbec and Merlot, than in Cabernet Sauvignon, and Thompson Seedless, which showed longer lesions and had larger vessel diameters. Symptomatic differences may also relate to water regimes that lead to vascular changes, as vessel dimensions change with water availability [18,53]. Thus, vessel size may correlate to susceptibility to GTDs, which could explain Cabernet Sauvignon’s greater susceptibility to isolates tested.

On the other hand, the susceptibility of cultivars could also be associated with inherent characteristics, such as lignin content and phenolic compounds, which play a significant role in resistance to these pathogens. Lignin, in particular, acts as a physical barrier that limits pathogen spread and disease progression [54,55]. Higher lignin content has been associated with greater resistance in cultivars like Merlot compared to Cabernet Sauvignon against GTD. Additionally, the same study reported higher levels of phenolic compounds in one-year-old wood infected with *Eutypa lata* from the Merlot cultivar, which was not observed in Cabernet Sauvignon. These characteristics may therefore influence the differential susceptibility observed between cultivars.

Given limited studies on Botryosphaeriaceae cross infection in Chile, this study provides a valuable foundation for epidemiological studies to support effective wood disease management strategies. This is the first study demonstrating that species of Botryosphaeriaceae from other hosts are potential sources of inoculum for vineyards in Central Chile, especially when they are grown in proximity, particularly for Cabernet Sauvignon, where *Neofusicoccum* species were more aggressive than other species of Botryosphaeriaceae.

The findings of this study regarding the aggressiveness of Botryosphaeriaceae species isolated from grapevines, apple trees, blueberries, and walnuts should be considered when designing appropriate sanitary measures for the prevention and management of GTD, both in fruit orchards and vineyards planted in close proximity. To effectively manage Botryosphaeria dieback in grapevines, producers must recognize the importance of proper sanitation practices, not only within the vineyards but also in surrounding plantations that serve as alternative hosts for these pathogens. Similarly, proper management of pruning residues is essential to prevent the spread of these pathogens. These residues should be removed from the vineyard and disposed of using methods such as burial, burning, or composting to prevent them from acting as inoculum reservoirs that could infect new wood lesions.

## Figures and Tables

**Figure 1 microorganisms-13-00331-f001:**
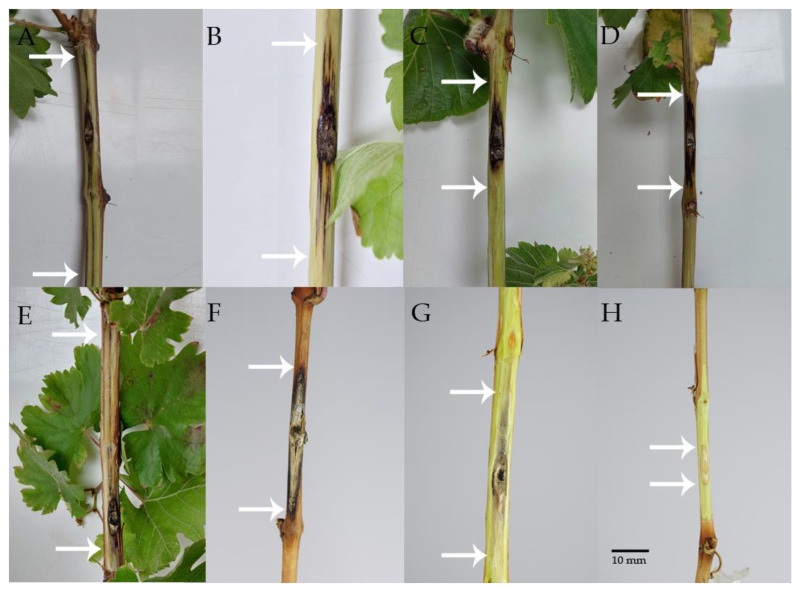
Internal and external necrotic lesions extending from wound sites in cuttings and green shoots of grapevines under greenhouse and vineyard conditions using the mycelial plug inoculation method. White arrows indicate the lesion boundaries. (**A**) Lesion caused by *N. parvum* (isolated from walnut) inoculated in cv. Cabernet Sauvignon in the greenhouse. (**B**) Lesion caused by *N. arbuti* (isolated from apple) inoculated in cv. Syrah in the greenhouse. (**C**) Lesion caused by *D. mutila* (isolated from walnut) inoculated in cv. Malbec in the greenhouse. (**D**) Lesion caused by *D. mutila* (isolated from apple) inoculated in cv. Sauvignon Blanc in the greenhouse. (**E**) Lesion caused by *N. parvum* (isolated from walnut) inoculated in cv. Aspirant Bouschet in the greenhouse. (**F**) External lesion caused by *N. parvum* (isolated from blueberry) inoculated in cv. Cabernet Sauvignon in the vineyard. (**G**) Lesion caused by *N. parvum* (isolated from grapevine) inoculated in cv. Merlot in the vineyard. (**H**) Negative control (non-inoculated green shoot of Cabernet Sauvignon in the vineyard) showed no damage (no canker and no vascular discoloration).

**Table 1 microorganisms-13-00331-t001:** Isolates of Botryosphaeriaceae spp. obtained from apples (*Malus domestica*), blueberries (*Vaccinium corymbosum*), grapevines (*Vitis vinifera*), and walnuts (*Juglans regia*) with dieback symptoms from the Maule Region, Central Chile, used in the present study.

Species/Isolate	Host	Isolate Code (Species-Host) ^w^	Locality (Latitude, Longitude) ^x^	Source
*Diplodia mutila*				
Bot-2017-DM2 ^yz^	Apple	Dm-apple	Talca (35°25′, 71°38′)	[33]
Dmnog-4 ^yz^	Walnut	Dm-wal1	Parral (36°08, 71°49)	[32]
Dmnog-3 ^y^	Walnut	Dm-wal2	Linares (35°50, 71°35)	[32]
Bot-2018-DM16-Ara ^y^	Blueberry	Dm-blue	Linares (35°50, 71°35)	This study
*D. seriata*				
Bot-2017-DS3 ^yz^	Apple	Ds-apple	Molina (35°50, 71°36)	[33]
Bot-2016-DS1-Vid ^yz^	Grapevine	Ds-vine	San Clemente (35°32, 71°29)	[24]
*Dothiorella sarmentotum * ^y^				
Dsar-2	Walnut	Dsar-wal	San Rafael (35°18′, 71°30)	[35]
*Lasiodiplodia theobromae * ^yz^				
Bot-2017-LT6	Apple	Lt-apple	Los Niches (35°08, 71°19)	[33]
*Neofusicoccum arbuti * ^yz^				
Bot-2018-NA32	Apple	Na-apple	Parral (36°08, 71°49)	[33]
*N. parvum*				
Bot-2016-NP10-Vid ^yz^	Grapevine	Np-vine	Curicó (34°58, 71°15)	[24]
Bot-2019-NP9-Ara ^yz^	Blueberry	Np-blue1	Linares (35°50, 71°35)	This study
Bot-2017-NP7-Ara ^z^	Blueberry	Np-blue2	Longaví (35°57, 71°41)	This study
Bot-2019-NP17-Nog ^yz^	Walnut	Np-wal	San Rafael (35°18, 71°30)	[35]

^w^ Isolate identification code given in this study. ^x^ Southern latitude and Western longitude. ^y^ Isolates of Botryosphaeriaceae inoculated with a mycelial plug on fresh wounds performed in the middle of cuttings (greenhouse) and green shoots (vineyard) of grapevines. ^z^ Isolates of Botryosphaeriaceae inoculated with a mycelial suspension (10^5^ fragments/mL) on fresh pruning wound performed in cuttings (greenhouse) and canes (vineyard) of grapevines.

**Table 2 microorganisms-13-00331-t002:** Mean necrotic lesion length (mm) of isolates of *Diplodia*, *Lasiodiplodia*, and *Neofusicoccum* on cuttings and canes of grapevine (*Vitis vinifera*) inoculated on pruning wounds with mycelial suspension (10^5^ fragments/mL) under greenhouse (four months) and vineyard (five months) conditions, respectively.

Isolate	Mean Necrotic Lengths (mm) on Cultivars of Grapevine ^y^					
	Cuttings in Greenhouse			Canes in the Vineyard		
	CS	SY	Mean	CS		
*Diplodia mutila*						
Dm-apple	19.1 e–i	12.5 i–l	15.8	64.9 bc		
Dm-wal1	30.1 a–c	13.9 h–k	22.0	84.6 a		
*D. seriata*						
Ds-apple	23.6 c–f	16.3 g–j	20.0	38.0 ef		
Ds-vine	25.3 b–e	14.9 g–j	20.1	42.6 de		
*Lasiodiplodia theobromae*						
Lt-apple	19.1 e–i	11.9 j–l	15.5	24.4 f		
*Neofusicoccum arbuti*						
Na-apple	31.3 ab	21.3 d–g	26.3	54.3 cd		
*N. parvum*						
Np-blue1	28.1 a–d	20.5 e–h	24.3	61.4 bc		
Np-blue2	25.5 b–e	16.9 f–j	21.2	73.4 ab		
Np-vine	34.7 a	21.4 d–g	28.0	52.5 cd		
Np-wal	29.8 a–c	19.8 e–h	24.8	86.2 a		
Control ^z^	7.6 kl	6.2 l	6.9	9.3 g		
Mean	24.9	16.0		53.8		
Analysis of variance	df	F	*p*	df	F	*p*
Cultivar (C)	1	230.9	<0.001	10	66.2	<0.001
Isolates (I)	10	37.5	<0.001	--	--	--
C × I	10	3.8	<0.001	--	--	--

^y^ Grapevine cultivars: CS = Cabernet Sauvignon, SY = Syrah. Means followed by the same letter did not differ significantly according to Tukey’s test (*p* < 0.05). Letter ranges include all intermediate letters. Lowercase letters indicate comparisons within the interaction between cultivar and isolate. ^z^ Negative control treatment inoculated using 40 µL of sterile distilled water.

**Table 3 microorganisms-13-00331-t003:** Necrotic lesion length (mm) of isolates of *Diplodia*, *Dothiorella*, *Lasiodiplodia*, and *Neofusicoccum* on cuttings and green shoots of grapevine (*Vitis vinifera*) inoculated in tissue wounds with mycelial plugs under greenhouse (six months) and vineyard (eight months) conditions, respectively.

Isolate	Mean Necrotic Lengths (mm) ^y^								
	Cuttings in Greenhouse						Green Shoots in the Vineyard		
	CS	SY	MA	SB	AB	Mean	CS	ME	Mean
*Diplodia mutila*									
Dm-apple	119.5 a–h	45.2 h–p	95.2 b–l	75.7 e–p	39.5 j–p	75.0	81.3 c–f	67.2 d–i	74.3
Dm-blue	26.8 o–r	82.7 c–n	65.8 e–p	51.2 h–p	34.5 m–p	52.2	74.0 c–g	57.1 f–k	65.5
Dm-wal2	43.8 i–p	54.7 f–p	47.9 g–p	39.6 k–p	63.8 e–p	50.0	66.0 d–i	44.0 h–l	55.0
Dm-wal1	34.4 k–p	54.5 h–p	31.6 m–p	38.0 l–p	36.2 l–p	38.9	75.2 c–g	47.0 i–l	61.1
*D. seriata*									
Ds-apple	50.6 f–p	24.8 p–s	23.8 p–s	42.5 i–p	134.9 a–f	55.3	55.9 f–k	31.3 lm	43.6
Ds-vine	112.6 a–i	37.2 k–p	48.3 h–p	32.6 n–p	32.8 m–p	52.7	58.0 e–k	42.0 kl	51.5
*Dothiorella sarmentorum*									
Dsar-wal	45.8 h–p	51.4 h–p	28.9 op	59.9 e–p	38.5 k–p	44.9	80.6 c–g	54.4 g–k	67.5
*Lasiodiplodia theobromae*									
Lt-apple	43.3 i–p	36.2 k–p	29.0 o–q	29.8 op	61.9 e–p	40.0	68.1 d–i	42.5 j–l	55.3
*Neofusicoccum arbuti*									
Na-apple	171.8 a–d	71.6 e–p	93.6 c–m	94.3 b–k	54.1 f–p	97.1	72.9 d–h	63.9 d–j	68.4
*N. parvum*									
Np-blue1	133.2 a–g	103.4 a–j	196.6 a–c	84.1 d–o	49.6 i–p	113.4	151.6 a	94.3 b–e	123.0
Np-vine	233.3 ab	100.5 b–k	86.9 b–l	142.2 a–g	66.1 d–p	125.8	95.9 a–d	114.2 a–c	105.0
Np-wal	253.7 a	93.4 b–k	69.3 d–o	76.0 d–o	174.9 a–e	133.5	134.2 ab	113.8 a–c	124.0
Control ^z^	10.0 q–t	9.6 r–t	6.9 t	8.3 t	9.0 st	8.8	15.0 n	16.6 mn	15.8
Mean	98.4	58.9	63.4	59.6	61.2		79.1	60.6	
Analysis of variance		df	F		*p*		df	F	*p*
Cultivar (C)		4	13.5		<0.001		1	73.0	<0.001
Isolates (I)		12	75.3		<0.001		12	64.9	<0.001
C × I		48	6.3		<0.001		12	4.0	<0.001

^y^ Grapevine cultivars: CS = Cabernet Sauvignon, SY = Syrah, MA = Malbec, SB = Sauvignon Blanc, AB = Aspirant Bouschet, and ME = Merlot. Means followed by the same letter did not differ significantly according to Tukey’s test (*p* < 0.05). Letter ranges include all intermediate letters. Lowercase letters indicate comparisons within the interaction between cultivar and isolate. ^z^ Negative control treatments were inoculated with PDA plug (5 mm diameter).

## Data Availability

The datasets used and/or analyzed during the current study are available from the corresponding author on request.

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
