# Peer review of "Host Jumps and Pathogenicity of Botryosphaeriaceae Species on Grapevines (*Vitis vinifera*) in Chile"

_microorganisms, 2025, doi:10.3390/microorganisms13020331_

Round 1

Reviewer 1 Report

Comments and Suggestions for Authors

The manuscript addressed the Pathogenicity of Botryosphaeriaceae Species from Diverse Hosts on Grapevines (Vitis vinifera) in Chile. The manuscript is generally good however, the following points need to be addressed:

1-  The title could changed to "Host Jumps and Pathogenicity of Botryosphaeriaceae Species on Grapevines (Vitis vinifera) in Chile" to be more concise and impactful.

2- In the abstract, specify the exact hosts from which the Botryosphaeriaceae isolates were obtained, and the type of grapevine cultivars used in the study.

3-Replace "various fruit and nut crops" with a more specific list of host plants. Specify the grapevine cultivars used in the study.

4- The introduction is quite broad, covering the general characteristics of Botryosphaeriaceae, their global distribution, and their impact on various crops. The specific focus of the study, which is to investigate cross-infection from other host species, is not clearly stated until the very end. The motivation and novelty of the study should be clearly mentioned. 

5- The method used to isolate the fungi from the infected plant material is not described.

6-Details such as tissue sampling (e.g., size of the sample, disinfection methods) and isolation techniques (e.g., single-spore isolation) are missing.

7- Details such as  size and shape of lesions, color changes in the wood (e.g., discoloration, darkening), and presence of cankers, gummosis should be given. 

8-The p value should be given to show the significant findings. Figure 1 should contain the magnification or a measurement scale and arrows to clearly demonstrate the lesions.

9-E laborate more on the discussion section and the following questions should be addressed in the relevent context:Are there any known differences in the susceptibility of Cabernet Sauvignon and Syrah to Botryosphaeriaceae infections? Could these differences be related to inherent cultivar characteristics (e.g., bark thickness, phenolic content) or cultural practices?

How might temperature, humidity, and other environmental factors influence the aggressiveness of different Botryosphaeriaceae species? How do the key findings  of this study advance our understanding of Botryosphaeriaceae diseases in grapevines?

10- Strengthen the conclusion by highlighting the significance of the main findings for understanding and managing Botryosphaeriaceae diseases in grapevines.

Author Response

For research article

Response to Reviewer 1 Comments

1. Summary

Thank you very much for taking the time to review this manuscript. Below you will find detailed responses and corresponding corrections written in red in the resubmitted file.

2. Questions for General Evaluation

Reviewer’s Evaluation

Response and Revisions

Does the introduction provide sufficient background and include all relevant references?

Must be improved

Introduction was better structured, and as recommended in comment 2, motivation and novelty of the study are now clearly mentioned. Specifications in the manuscript are written in red and detailed in the point-by-point response to comment 2.

Is the research design appropriate?

Yes

Are the methods adequately described?

Can be improved

Methods section was improved by adding the suggested information from comments 5 and 6. Specifications in the manuscript are written in red and detailed in the point-by-point response to these comments.

Are the results clearly presented?

Can be improved

To improve the clarity of the results and discussion, changes were made in the manuscript as per recommended in comments 7 to 9. Specifications in the manuscript are written in red and detailed in the point-by-point response to related comments.

Are the conclusions supported by the results?

Can be improved

Additional information highlighting the significance of the main findings were added to conclusions, as recommended in comment 10. Specifications in the manuscript are written in red and detailed in the point-by-point response to comment 10.

3. Point-by-point response to Comments and Suggestions for Authors

Comments 1: The title could changed to "Host Jumps and Pathogenicity of Botryosphaeriaceae Species on Grapevines (Vitis vinifera) in Chile" to be more concise and impactful.

Response 1: Thank you very much for your suggestion. It has been taken into account, and the title has been changed. The change can be found on page 1, lines 2 to 3, written in red.

Comments 2:  In the abstract, specify the exact hosts from which the Botryosphaeriaceae isolates were obtained, and the type of grapevine cultivars used in the study.

Response 2: Thank you very much for your comment. It has been taken into account, and the hosts from which the Botryosphaeriaceae isolates were obtained have been added, as well as the cultivars used in the study. The change can be found in the abstract on page 1, lines 15 and 17-18, written in red.

Comments 3: Replace "various fruit and nut crops" with a more specific list of host plants. Specify the grapevine cultivars used in the study.

Response 3: Thank you very much for your comment. Given the word limit for the abstract, the extensive list of hosts in which Botryosphaeriaceae species have been described and identified in Chile, along with the additional details added to improve precision as recommended in comment 2, it was decided to maintain the generic expression in the abstract, while ensuring the clarity of this information in the introduction. The details regarding the host list can be found in the introduction on page 2, paragraph 2, lines 54 and 55.

Regarding the cultivars used, they have been added in response to your comment, both in the abstract (page 1, lines 17 and 18) and in the materials and methods section (page 3, paragraph 1, lines 83 – 84 and paragraph 3, lines 99 – 100).

Comments 4: The introduction is quite broad, covering the general characteristics of Botryosphaeriaceae, their global distribution, and their impact on various crops. The specific focus of the study, which is to investigate cross-infection from other host species, is not clearly stated until the very end. The motivation and novelty of the study should be clearly mentioned. 

Response 4: Thank you for your comment. It has been taken into account. A new paragraph has been added in the introduction to clearly outline the motivation and novelty of this study (page 2, paragraph 3, lines 60 to 67, written in red). Regarding the specific focus of the study, we chose to present the background in a structured manner, progressing from the more general to the more specific. As such, the aim of the article is stated at the end of the introduction to provide a logical flow and context for the study.

Comments 5: The method used to isolate the fungi from the infected plant material is not described.

Response 5: Thank you for your comment. We understand that your inquiry refers to the methods used to isolate the pathogens from the infected plant material during the evaluation process. To clarify the description of the method used for pathogen isolation from infected tissue, we have decided to provide a separate section in the Materials and Methods that specifically addresses the re-isolation procedure. These changes can be found on page 4, lines 149.

Comments 6: Details such as tissue sampling (e.g., size of the sample, disinfection methods) and isolation techniques (e.g., single-spore isolation) are missing.

Response 6: Thank you for your comment. Below are the locations in the manuscript where the mentioned details are described: Sample size: page 4, line 153; Disinfection methods: page 4, line 151-152.

Comments 7: Details such as size and shape of lesions, color changes in the wood (e.g., discoloration, darkening), and presence of cankers, gummosis should be given.

Response 7: Thank you for your comment. It has been taken into account. To ensure a clear description of the lesions, these details have been specified at the beginning of the results section for each inoculation method. The changes can be found on page 5, paragraph 4, lines 177-178, and on page 7, paragraph 1, lines 233 to 235, written in red.

Comments 8: The p value should be given to show the significant findings. Figure 1 should contain the magnification or a measurement scale and arrows to clearly demonstrate the lesions.

Response 8: Thank you for your comment. Each panel in Figure 1 represents a lesion associated with a specific isolate. Therefore, the significant contrasts are obtained through post-hoc tests, which may reveal multiple significant differences between isolates. For this reason, the figure is intended to visually represent the characteristics of the lesions caused by different isolates across the various cultivars and conditions used, with these differences further detailed in the tables. Regarding the Figure, both the measurement scale and arrows have been added to clarify the lesions. The changes can be found page 7, lines 236-237.

Comments 9: Elaborate more on the discussion section and the following questions should be addressed in the relevant context: Are there any known differences in the susceptibility of Cabernet Sauvignon and Syrah to Botryosphaeriaceae infections? Could these differences be related to inherent cultivar characteristics (e.g., bark thickness, phenolic content) or cultural practices?

How might temperature, humidity, and other environmental factors influence the aggressiveness of different Botryosphaeriaceae species? How do the key findings of this study advance our understanding of Botryosphaeriaceae diseases in grapevines?

Response 9: Thank you very much for your comments and queries. They have been taken into account and have contributed to enriching the discussion. The associated changes can be found on page 11, paragraph 5, lines 414 to 419 and page 12, paragraph 4, lines 442 to 450, written in red.

Comments 10: Strengthen the conclusion by highlighting the significance of the main findings for understanding and managing Botryosphaeriaceae diseases in grapevines.

Response 10: Thank you for your suggestion. A paragraph has been added to the conclusions highlighting the significance of the main findings for understanding and managing Botryosphaeriaceae diseases in grapevines. Additionally, in response to comments from the other reviewer, further details on preventive strategies have been included in the conclusions. These changes can be found on page 12, paragraph 6, lines 458 to 468, written in red.

Reviewer 2 Report

Comments and Suggestions for Authors

The manuscript with the title “Cross-infection of Botryosphaeriaceae species obtained from different fruit tree hosts causing dieback on grapevines (Vitis vinifera) in Chile” presents a comparative study on infectivity of some species from the Botryosphaeriaceae on grapevine, that were isolated from other fruit crops, identifying their phytopatological potential on vine varieties.

The Introduction provides a good motivation for the research.

Results. I suggest that perhaps when comparing the treatments in the text, to expresses the differences relative (%) of each other, or to control because it would be easier to understand how large differences are.

Tables. I am sorry to ask this question, but it is not entirely obvious to me what the lines between lettercase significance represent?

Discussion. Authors provide a good comparison with data from specialty literature on the topic. However, what recommendations do they suggest or propose? Can there be named or identified some preventive strategies for Chile vineyards?

General remark. Overall, the study reported is relevant because epidemiological studies such as these support the development of effective wood disease prevention and management strategies.

Best regards.

Author Response

For research article

Response to Reviewer 2 Comments

1. Summary

Thank you very much for taking the time to review this manuscript. Below you will find detailed responses and corresponding corrections written in red in the resubmitted file.

2. Questions for General Evaluation

Reviewer’s Evaluation

Response and Revisions

Does the introduction provide sufficient background and include all relevant references?

Yes

Is the research design appropriate?

Yes

Are the methods adequately described?

Yes

Are the results clearly presented?

Can be improved

To improve the clarity of the results and in response to comment 4, additional information was included to facilitate the interpretation of the specific results for each isolate in the tables. Specifications in the manuscript are written in red and detailed in the point-by-point response to comment 4.

Are the conclusions supported by the results?

Can be improved

Additional information addressing preventive strategies has been incorporated into the conclusions, as suggested in comment 5. Specifications in the manuscript are written in red and detailed in the point-by-point response to comment 5.

3. Point-by-point response to Comments and Suggestions for Authors

Comments 1: The manuscript with the title “Cross-infection of Botryosphaeriaceae species obtained from different fruit tree hosts causing dieback on grapevines (Vitis vinifera) in Chile” presents a comparative study on infectivity of some species from the Botryosphaeriaceae on grapevine, that were isolated from other fruit crops, identifying their phytopatological potential on vine varieties.

Response 1: Thank you for the accurate summary, which captures the relevance of our study on Botryosphaeriaceae species and their impact on vine varieties.

Comments 2: The Introduction provides a good motivation for the research.

Response 2: Thank you for your positive feedback.

Comments 3: Results. I suggest that perhaps when comparing the treatments in the text, to expresses the differences relative (%) of each other, or to control because it would be easier to understand how large differences are.

Response 3: We appreciate the suggestion to express the relative differences (%) to facilitate interpretation. However, we believe that the significant differences among treatments are already clearly represented through the statistical significance letters derived from Tukey’s tests, which provide a simple and standardized way to identify which treatments differ significantly from one another.

Additionally, including relative differences could complicate the description of the results. For example, expressing the relative differences for each treatment compared to the corresponding control would require phrasing each comparison as follows: "Treatment A showed a 78.03% increase compared to the control, while Treatment B showed a 67.11% increase, and Treatment C a 50.27% increase." While this level of detail is accurate, it serves a similar purpose to the statistical significance letters from Tukey's test, which already summarize this information effectively and accessibly (e.g., treatments associated with the letter "a" are more different from those associated with "e" than from those associated with "b").

Comments 4: Tables. I am sorry to ask this question, but it is not entirely obvious to me what the lines between lettercase significance represent?

Response 4: In response to your question, the lines between lettercase significance represent the range of letters associated with the results of the post-hoc tests. Since many treatments shared the same letters, we decided to use this line to simplify the presentation of the information. For example, if a treatment has seven letters associated with it (a, b, c, d, e, f, g, and h), it would be represented as "a – h". This approach helps to clarify which treatments are statistically similar, as treatments sharing the same range of letters are not significantly different from each other, aligned with the response to previous comment. To ensure clarity regarding the meaning of the lettercase in the tables, the specifications were added in the footnotes of each table. These can be found in Table 2, page 6, line 200 - 201 and Table 3, page 9, line 283, written in red.

Comments 5: Discussion. Authors provide a good comparison with data from specialty literature on the topic. However, what recommendations do they suggest or propose? Can there be named or identified some preventive strategies for Chile vineyards?

Response 5: Thank you very much for your comment. In response, a paragraph mentioning preventive strategies has been added to the conclusions. This paragraph can be found on page 12, paragraph 6, lines 458 to 468, written in red.

Comments 6: General remark. Overall, the study reported is relevant because epidemiological studies such as these support the development of effective wood disease prevention and management strategies

Response 6: Thank you for your positive feedback. We appreciate your recognition of the study's relevance.